# ILB^®^, a Low Molecular Weight Dextran Sulphate, Restores Glutamate Homeostasis, Amino Acid Metabolism and Neurocognitive Functions in a Rat Model of Severe Traumatic Brain Injury

**DOI:** 10.3390/ijms23158460

**Published:** 2022-07-30

**Authors:** Giacomo Lazzarino, Valentina Di Pietro, Marco Rinaudo, Zsuzsanna Nagy, Nicholas M. Barnes, Lars Bruce, Stefano Signoretti, Renata Mangione, Miriam Wissam Saab, Barbara Tavazzi, Antonio Belli, Giuseppe Lazzarino, Angela Maria Amorini, Ann Logan

**Affiliations:** 1Departmental Faculty of Medicine and Surgery, UniCamillus—Saint Camillus International University of Health and Medical Sciences, Via di Sant’Alessandro 8, 00131 Rome, Italy; giacomo.lazzarino@unicamillus.org; 2Neurotrauma and Ophthalmology Research Group, School of Clinical and Experimental Medicine, College of Medical and Dental Sciences, University of Birmingham, Edgbaston, Birmingham B15 2TT, UK; v.dipietro@bham.ac.uk (V.D.P.); a.belli@bham.ac.uk (A.B.); 3National Institute for Health Research, Surgical Reconstruction and Microbiology Research Centre, Queen Elizabeth Hospital, Edgbaston, Birmingham B15 2TH, UK; n.m.barnes@bham.ac.uk; 4Department of Neuroscience, Catholic University of the Sacred Heart of Rome, Largo F. Vito 1, 00168 Rome, Italy; marco.rinaudo@unicatt.it; 5Fondazione Policlinico Universitario A. Gemelli IRCCS, Largo A. Gemelli 8, 00168 Rome, Italy; 6Institute of Clinical Sciences, College of Medical and Dental Sciences, University of Birmingham, Edgbaston, Birmingham B15 2TT, UK; z.nagy@bham.ac.uk; 7Tikomed AB, 26303 Viken, Sweden; lars.bruce@tikomed.com; 8Division of Neurosurgery, Department of Emergency and Urgency, S. Eugenio/CTO Hospital, A.S.L. Roma 2, Piazzale dell’Umanesimo 10, 00144 Rome, Italy; 9Department of Basic Biotechnological Sciences, Intensive and Perioperative Clinics, Catholic University of the Sacred Heart of Rome, Largo F. Vito 1, 00168 Rome, Italy; renata.mangione@unicatt.it; 10Division of Medical Biochemistry, Department of Biomedical and Biotechnological Sciences, University of Catania, Via S. Sofia 97, 95123 Catania, Italy; miriam.saab@phd.unict.it (M.W.S.); lazzarig@unict.it (G.L.); amorini@unict.it (A.M.A.); 11Department of Biomedical Sciences, University of Warwick, Coventry CV4 7HL, UK; ann.logan@warwick.ac.uk; 12Axolotl Consulting Ltd., Droitwich WR9 0JS, UK

**Keywords:** severe traumatic brain injury, low molecular weight dextran sulphate, amino acids, glutamate excitotoxicity, neurocognitive functions, oxidative/nitrosative stress, methyl cycle, HPLC

## Abstract

In a previous study, we found that administration of ILB^®^, a new low molecular weight dextran sulphate, significantly improved mitochondrial functions and energy metabolism, as well as decreased oxidative/nitrosative stress, of brain tissue of rats exposed to severe traumatic brain injury (sTBI), induced by the closed-head weight-drop model of diffused TBI. Using aliquots of deproteinized brain tissue of the same animals of this former study, we here determined the concentrations of 24 amino acids of control rats, untreated sTBI rats (sacrificed at 2 and 7 days post-injury) and sTBI rats receiving a subcutaneous ILB^®^ administration (at the dose levels of 1, 5 and 15 mg/kg b.w.) 30 min post-impact (sacrificed at 2 and 7 days post-injury). Additionally, in a different set of experiments, new groups of control rats, untreated sTBI rats and ILB^®^-treated rats (administered 30 min after sTBI at the dose levels of 1 or 5 mg/kg b.w.) were studied for their neurocognitive functions (anxiety, locomotor capacities, short- and long-term memory) at 7 days after the induction of sTBI. Compared to untreated sTBI animals, ILB^®^ significantly decreased whole brain glutamate (normalizing the glutamate/glutamine ratio), glycine, serine and γ-aminobutyric acid. Furthermore, ILB^®^ administration restored arginine metabolism (preventing nitrosative stress), levels of amino acids involved in methylation reactions (methionine, L-cystathionine, S-adenosylhomocysteine), and N-acetylaspartate homeostasis. The macroscopic evidences of the beneficial effects on brain metabolism induced by ILB^®^ were the relevant improvement in neurocognitive functions of the group of animals treated with ILB^®^ 5 mg/kg b.w., compared to the marked cognitive decline measured in untreated sTBI animals. These results demonstrate that ILB^®^ administration 30 min after sTBI prevents glutamate excitotoxicity and normalizes levels of amino acids involved in crucial brain metabolic functions. The ameliorations of amino acid metabolism, mitochondrial functions and energy metabolism in ILB^®^-treated rats exposed to sTBI produced significant improvement in neurocognitive functions, reinforcing the concept that ILB^®^ is a new effective therapeutic tool for the treatment of sTBI, worth being tested in the clinical setting.

## 1. Introduction

Traumatic brain injury (TBI) is a heterogeneous and complex pathology occurring in the most complex organ of the body. The US Centers for Disease Control and Prevention defined TBI as a “disruption in the normal function of the brain that can be caused by a bump, blow or jolt to the head, or penetrating injury” [1]. Independently of sex, age and a patient’s social status, TBI affects more than 10 million people/year worldwide, representing 30% to 40% of all injury-related mortalities. In the near future (within 2030), epidemiological forecasts suggest that the number of patients affected by TBI-related disabilities will be 2–3 times higher than the sum of those affected by the most frequent chronic neurodegenerations and cerebrovascular disorders. A vast number of traumatic events can cause TBI, which, therefore, is characterized by profound heterogeneity, rendering difficult the comprehension of its pathobiological mechanisms and the prediction of the clinical evolution of TBI patients, including their outcome. The common feature among all TBI is that part of the energy associated with the traumatic event (primary injury) is absorbed by the brain tissue, invariably triggering a plethora of biochemical, metabolic, and molecular alterations (secondary injury), profoundly affecting various biological processes and functions of neuronal cells [2,3]. Among them, glutamate (Glu) excitotoxicity [4,5,6,7] and alterations of the brain levels of numerous free amino acids (FAA) involved in relevant biochemical activities [8,9,10,11,12,13,14,15] have been reported to occur following TBI. For these reasons, TBI patients are still in search of valid pharmacological treatments that can ameliorate their outcome.

In a well-established experimental model of graded TBI in the rat, we have shown that severe TBI (sTBI) produced long-lasting glucose dysmetabolism with energy crisis [16,17], imbalance of mitochondrial dynamics and functions [18,19], sustained oxidative/nitrosative stress [20,21] and alterations of NAA homeostasis [22]. Furthermore, concentrations of Glu and other FAA underwent significant changes, corroborating the occurrence of Glu excitotoxicity following sTBI [23]. Very recently, using the same animal model of sTBI, we found that the subcutaneous administration of a new low-molecular weight dextran sulphate (ILB^®^) following sTBI is able to significantly improve brain energy metabolism and mitochondrial phosphorylating capacity and to decrease oxidative/nitrosative stress [24].

In the present study, using the brain extracts of the same sTBI animals from the aforementioned ILB^®^ study, we evaluated in more detail the effects of this drug administration on Glu excitotoxicity, FAA and NAA brain concentrations at different times after sTBI. In addition, in a different group of animals, we also evaluated whether ILB^®^ treatment may positively influence the adverse alterations of neurocognitive functions induced by sTBI.

## 2. Results

### 2.1. Effect of ILB^®^ on Glutamate Excitotoxicity of the Post-sTBI Brain

Compared to untreated intact controls (Figure 1), significantly higher values of Glu (A), Glu + Gln (C) and Glu/Gln ratio (D) were observed in untreated sTBI rats, at both times after trauma (*q* < 0.05), whilst Gln (B) was increased only at 2 days after injury (*q* < 0.05). The administration of ILB^®^ 30 min after sTBI had variable effects on the aforementioned parameters, depending on both the dose and the time after impact. In particular, compared to untreated sTBI rats, the 1 mg/kg b.w. ILB^®^ dose did not produce significant protection against the sTBI-induced alterations of Glu and Gln homeostasis. Differently, the administration of both 5 and 15 mg/kg b.w. ILB^®^ significantly decreased the concentrations of Glu, Gln and Glu + Gln at both times post-sTBI (*q* < 0.05), with the highest ILB^®^ dose demonstrating the most evident effects on Glu, Glu + Gln and Glu/Gln ratio at both time points after trauma (*q* < 0.05 compared to the corresponding time of untreated sTBI rats; not significant, compared to intact controls).

The concomitant increase in the mean cerebral concentrations of both Gly and Ser, the two most effective amino acids acting as positive neuromodulators of the glutamate ionotropic NMDA receptor, in rats without and with the s.c. administration of increasing ILB^®^ doses at 2 and 7 days post-impact, allows to better evaluate the extent of Glu excitotoxicity seen following sTBI (Figure 2).

Compared to intact controls, cerebral Gly levels (**A**) showed a significant increase at 2 days only (*q* < 0.001), whilst Ser (**B**) increased either at 2 or 7 days post-impact (*q* < 0.03 and *q* < 0.001, respectively). To counteract the excitotoxic effects caused by excess Glu and by FAA stimulating Glu receptors (Gly and Ser), brain extracts of untreated sTBI rats had higher levels of GABA (**C**) at both times after injury (*q* < 0.002 and *q* < 0.001, respectively). Treatments with increasing doses of ILB^®^ affected, only in part, the alterations of Gly and Ser seen after sTBI. No differences were observed when comparing the Gly values of ILB^®^-treated rats with those of untreated sTBI animals, with the exception of rats administered with 15 mg/kg b.w. ILB^®^ at 7 days post-impact (*q* < 0.001). By contrast, at 2 days post-injury, the beneficial effects of ILB^®^ treatment produced a dose-dependent decrease in the cerebral concentrations of Ser, which were significantly lower than those measured in untreated sTBI rats at the same time point (*q* < 0.01 and *q* < 0.001, respectively). At longer times post-impact (7 days), no differences between treated and untreated animals were observed, even in rats administered with the highest dose of ILB^®^ tested (15 mg/kg b.w.) which had Ser values not significantly different from those found in intact control animals.

The decrease in the excitotoxic conditions seen in ILB^®^-treated rats was corroborated by the lower GABA concentrations also measured in these animals, compared to those detected in untreated-sTBI rats. A dose-dependent effect of the drug was clearly observable at 7 days post-sTBI, when animals receiving the 1 mg/kg b.w. ILB^®^ dose had values not different from those of untreated-sTBI rats and higher than those of controls (*q* < 0.005), whilst those treated with 5 or 15 mg/kg b.w. ILB^®^ had lower GABA concentrations than those of untreated-sTBI animals (*q* < 0.05 and *q* < 0.005, respectively, for the 5 mg/kg b.w. ILB^®^ dose; *q* < 0.005 and *q* < 0.002, respectively, for the 15 mg/kg b.w. ILB^®^ dose), cerebral levels that were not different from those measured in intact controls.

### 2.2. Effect of ILB^®^ on Nitrosative Stress and Metabolism of Met, Tau and Trp

Figure 3 demonstrates that sTBI induced a remarkable imbalance in Arg metabolism, leading to changes in the cerebral levels either of Arg (A) or of its metabolites Orn (B) and Citr (C), with an overall dramatic decrease in the global arginine bioavailability ratio (GABR = Arg/Orn + Citr) (D). At both times post-impact, this parameter, indirectly measuring nitrosative stress due to excess nitric oxide production [25,26], was significantly lower than the value measured in intact controls (*q* < 0.001).

The administration of ILB^®^, by increasing the cerebral levels of Arg and Orn and concomitantly decreasing that of Citr, produced a dose-dependent significant amelioration of the GABR at both 2 and 7 days post-injury (*q* < 0.01, compared to the corresponding times of untreated sTBI animals), with the 5 and 15 mg/kg b.w. ILB^®^ doses restoring this parameter to the level of that measured in intact control rats.

Figure 4 shows that untreated rats experiencing sTBI had a profound imbalance in Met homeostasis, evidenced by the decrease in its concentrations (A) at both times after injury (compared to intact controls, *q* < 0.02 and *q* < 0.05 at 2 and 7 days post-sTBI, respectively) and a concomitant increase in its catabolites L-Cystat (B) and SAH (C) (compared to intact controls, *q* < 0.005 at 2 and 7 days post-sTBI, for both compounds). Treatment with ILB^®^ at 30 min post-sTBI produced a dose-dependent increase in Met concentrations up to values similar to those of intact controls and a decrease in both Met catabolites, particularly evident in the case of SAH.

Following sTBI, control untreated rats showed significant increases in the cerebral concentrations of Tau (D), at both 2 (*q* < 0.005) and 7 days (*q* < 0.03) after trauma. ILB^®^ effectively decreased Tau levels only at 7 days post-impact, with the 5 and 15 mg/kg b.w. doses producing significant decreases compared to untreated sTBI animals (*q* < 0.05). Severe TBI also caused profound Trp (E) depletion (*q* < 0.002 and *q* < 0.001 at 2 and 7 days post-injury, respectively), which was effectively rescued by each of the three doses of ILB^®^ tested.

### 2.3. Effect of ILB^®^ on NAA Metabolism

Figure 5 demonstrates that untreated animals receiving sTBI experienced a significant increase in cerebral Asp levels (A) (*q* < 0.001 and *q* < 0.006 at 2 and 7 days post-injury, respectively), which was accompanied by a dramatic depletion of NAA (B) (*q* < 0.001 at both 2 and 7 days post-injury).

Cerebral Asp levels were only partly decreased by the 5 mg/kg b.w. ILB^®^ dose at 7 days post-injury (not significant compared to either intact controls or untreated sTBI rats), but the levels were more effectively reduced by the 15 mg/kg b.w. ILB^®^ dose at both times after trauma (compared to untreated sTBI animals, *q* < 0.001 at both 2 and 7 days post-injury; levels not significantly different at both time points compared to intact controls). Notwithstanding the modest effects on Asp variation, ILB^®^ treatment provoked a clear dose-dependent increase in NAA brain levels that was particularly evident at 7 days when the highest dose of ILB^®^ tested restored NAA concentrations to values not significantly different from those observed in intact controls.

### 2.4. Effect of ILB^®^ on Neurocognitive Functions of sTBI Rats: The OFT Test for Anxiety

As previously indicated, the OFT test animals were allowed to explore a 90 cm × 90 cm arena for 10 min on day 1 after sTBI or surgical procedures. The longer the time spent in the middle square of the test arena (45 cm × 45 cm) the less the anxiety experienced by the animal. As shown in the heat maps (Figure 6) and quantified in Figure 7, both groups of untreated sTBI and sTBI + 1 mg/kg b.w. ILB^®^-treated animals spent more time in the border zone (the area between the external border and the middle square) of the arena and less time in the center zone, compared to intact controls and sTBI + 5 mg/kg b.w. ILB^®^-treated animals.

The quantification of the total time spent by animals of each treatment group in the center zone of the test arena or in the border zone is illustrated in Figure 7. To exclude a preference for a specific corner of the arena, we preliminarily quantified the time spent by animals of each group in each corner, finding no significant differences between groups (data not shown).

Whilst untreated sTBI and sTBI + 1 mg/kg b.w. ILB^®^-treated rats spent a significantly lower number of seconds in the center of the arena (A) than did those in the intact control group (*q* < 0.001 controls versus untreated sTBI; *q* < 0.05 controls versus sTBI + 1 mg/kg b.w. ILB^®^), sTBI + 5 mg/kg b.w. ILB^®^-treated animals did not differ from intact control rats in their permanence in the center of the field. As a consequence of the shorter time spent in the center zone, both untreated sTBI and sTBI + 1 mg/kg b.w. ILB^®^-treated rats spent more time in the border zone of the arena (B) (*q* < 0.001 in untreated sTBI versus intact controls; *q* < 0.05 sTBI + 1 mg/kg b.w. ILB^®^ versus intact controls) because of their higher levels of anxiety. The decrease in anxiety induced by the administration of 5 mg/kg b.w. ILB^®^ 30 min after sTBI was confirmed by the significant difference in the time spent in the border zone of the test arena (q < 0.01 compared to untreated sTBI rats). Therefore, this group of animals had levels of anxiety that were not significantly different from those of intact control rats, either when measuring the time spent in the center or in the border zone of the test arena. No differences were observed when comparing the four groups of rats for the distance walked during the OFT test (C), suggesting no locomotor impairment in any of these animals.

### 2.5. Effect of ILB^®^ on Neurocognitive Functions of sTBI Rats: The NOR-STM and NOR-LTM Tests

As explained in the methods section relating to rat exploratory responses to objects placed in the test arena, measuring time to object exposition, the duration of the test and the interval between training and test allowed evaluation of short-term and long-term memory by assigning a Preference Index (PI) to their behavior. As shown in Figure 8, no significant differences in the PI values for the NOR-STM test (A), and, therefore, in short-term memory, were recorded in any of the groups of animals.

In the NOR-LTM test, animals were exposed for 10 min to a pair of identical objects placed within the same arena of the OFT test (training phase). Twenty-four hours later, two objects were again placed into the arena, with one of the two objects changed (test phase). The time spent exploring the two objects was recorded, and the preference for the novel object was calculated as the ratio between the time spent exploring the novel object and the time spent exploring both the novel and the familiar objects, multiplied by 100. Values of this Preference Index (PI) close to 50% suggested no discrimination, while PI values higher than 60% were considered as indicative of the novel object preference and, therefore, of intact object memory.

The results of NOR-LTM tests (B) clearly show the decrease in this neurocognitive indicator of long-term memory in the groups of untreated rats experiencing sTBI as well as in those rats receiving 1 mg/kg b.w. ILB^®^ 30 min after sTBI (*q* < 0.003 comparing each of these two groups with intact controls). Again, the administration of 5 mg/kg b.w. ILB^®^ 30 min after injury produced a recovery of this neurocognitive function, as evidenced by the lack of significant differences in the comparison with sham operated controls and in the significant difference in the comparison with the group of sTBI-only rats (*q* < 0.005).

### 2.6. Effect of ILB^®^ on Selected Brain Metabolites Measured in the Neurocognitive Study Rats

Table 1 summarizes the effects of ILB^®^ administration on the cerebral concentrations of ATP, GTP, NAD^+^, ascorbate, GSH, NAA, Glu, GABR and Met measured in the brain extracts of the same animals studied to assess their neurocognitive functions. In comparison with the values determined in untreated sTBI rats, the administration of ILB^®^ at 5 mg/kg produced a significant increase in all metabolic parameters measured except for Met. Notwithstanding this observation, significant differences were observed when the comparison was made with the corresponding values determined in the group of sham-operated control rats.

## 3. Discussion

In a study recently published by our research group, we have shown that the s.c. administration of ILB^®^, 30 min after injury, dose-dependently ameliorates cerebral energy metabolism, mitochondrial functions, antioxidant defenses and oxidative/nitrosative stress in rats experiencing diffused sTBI [24]. In the present study, using the same protein-free brain extracts analyzed in the aforementioned research, we provide additional evidence demonstrating that ILB^®^ treatment prevents Glu-mediated excitotoxicity, decreases the excess of nitric oxide production, restores Met homeostasis, normalizes Tau and Trp concentrations and repristinates NAA homeostasis. Equally important is the finding that the ILB^®^-mediated improvement of brain metabolism is clearly mirrored by significant ameliorations in sTBI-induced alterations of fundamental neurocognitive functions.

The analysis of FAA in brain extracts of untreated sTBI rats confirmed that both Glu and the activators of NMDA ionotropic Glu receptors (Gly and Ser) increased at short (2 days) and longer times (7 days) post-impact. These changes also involved Gln, leading to imbalance of the Glu-Gln cycle between astrocytes and neurons, an essential process that prevents Glu-mediated excitotoxicity through the efficient Glu removal from the extracellular space, [27], and were accompanied by an increase in GABA levels to counteract the excitatory stimuli [23]. The s.c. administration of ILB^®^ at 30 min after sTBI, by normalizing cerebral Glu and Gln concentrations (as well as those of the Glu + Gln sum and of the Glu/Gln ratio), effectively decreased Glu-mediated excitotoxicity and the imbalance of the Glu-Gln cycle. These effects are certainly linked to the improved energy metabolism and mitochondrial functions previously observed in brain extracts of the same animals [24]. It is well demonstrated that TBI-induced calcium overload disrupts cellular bioenergetics, due to massive calcium entry into the mitochondrial compartment [28,29] that leads to malfunctioning of the electron transport chain (ETC) linked to oxidative phosphorylation (OXPHOS), with consequent energy penalty [16,17,18,19,20,21,22,30,31,32].

In our experiments, Glu-mediated excitotoxicity in rats experiencing sTBI was escorted by an imbalance of Arg metabolism, leading to alterations of Orn and Citr concentrations and significant changes in the GABR index. As suggested by previous observations showing increases in nitrate and nitrite (as stable end-products of NO metabolism) following sTBI [24], changes in the aforementioned parameters are probably caused by an increase in NO production, strongly corroborating the evidence of sustained nitrosative stress concomitant to Glu excitotoxicity in the TBI-injured brain [20,33]. Treating sTBI animals with increasing ILB^®^ concentrations produced beneficial effects on Arg metabolism, leading to a decrease in Citr, an increase in Orn concentrations and normalization of the GABR index, with consequent diminution of NO production and reduction of the risks of nitrosative stress insurgence [24]. It is worth underlining that the effects of ILB^®^ towards NO overproduction and nitrosative stress have recently been found even in the clinical setting, in a cohort of ALS patients in whom ILB^®^ administration significantly decreased the circulating levels of nitrite and nitrate [34].

The metabolic damages caused by sTBI produced a remarkable imbalance in the so-called methyl cycle [35], involving a decrease in brain levels of Met and a concomitant increase in L-Cystat and SAH, that was effectively counteracted by the treatment with ILB^®^. The tendency to restore the correct concentrations of Met, L-Cystat and SAH in the post-sTBI brain of ILB^®^-treated rats is of particular relevance because of the involvement of these compounds in various crucial biochemical processes of cerebral cells. For instance, it has been demonstrated that alterations of the methyl cycle modify the level of mitochondrial DNA methylation following repeat mild TBI, playing a central role in decreasing mitochondrial biogenesis and protracting mitochondrial malfunctioning [36]. In rats experiencing TBI induced by repeated blast exposure, it was shown that the decrease in brain Met significantly affected the cerebral concentrations of GSH [37] and, furthermore, decreased circulating levels of Met were found in the blood of sTBI patients [38], thus corroborating the importance of the beneficial effects on brain Met displayed by ILB^®^ administration in our experiments.

The drug treatment also diminished the sTBI-mediated increase in brain L-Cystat and normalized the level of SAH. Although obtained in a fish model of TBI, it was found that head trauma causes an increase in the expression of the L-Cystat synthesizing enzyme (L-cystathione-β-synthase) in the post-injured brain [39], thereby possibly explaining the 1.9 and 1.7-fold increase in L-Cystat we found in sTBI rats at 2 and 7 days post-impact, respectively. On the other hand, since the precursor of GSH biosynthesis (L-cysteine) originates from the enzymatic scission of L-Cystat, the increased levels of L-Cystat following sTBI may be causative of a decreased rate of GSH biosynthesis due to decreased L-cysteine availability. These phenomena, in combination with oxidative stress-mediated GSH oxidation [21], cause a net depletion of cerebral GSH [24,37,40], thus exposing the brain to higher risk of ROS-mediated damage to biomolecules that was successfully counteracted by ILB^®^ treatment.

The last evident, but no less important, effect fulfilled by ILB^®^ administration was the recovery of cerebral NAA concentrations, with the concomitant normalization of brain Asp. As previously shown, NAA is a reliable surrogate index of mitochondrial energy metabolism, strictly connected to ATP availability and cell energy state [19,23,41]. The homeostasis of this molecule is deeply imbalanced by TBI because of changes in the expressions of the two genes controlling NAA biosynthesis (NAT8L) and degradation (aspartoacylase, ASPA) [22]. In brain extracts of the same animals we used in the current study, we found that ILB^®^ administration rescued energy metabolism and mitochondrial functions [24], thus explaining the effects of the drug on NAA and Asp. These findings may have a relevant implication for future clinical trials of ILB^®^ effects in TBI patients, since NAA can easily be measured in vivo using proton magnetic resonance spectroscopy (^1^H-MRS) [42] and since ^1^H-MRS quantification clearly demonstrated significant NAA depletion in the brains of TBI patients [43,44,45,46,47,48,49].

The very relevant finding of this research is the strict correlation found when comparing the effects on neurocognitive function, performed in a different set of experiments comparing sham-operated control rats with rats experiencing sTBI, without and with the administration of 1 or 5 mg/kg b.w. ILB^®^ given 30 min post-injury, with the results for the brain levels of selected metabolites that are representative of energy metabolism, antioxidant defenses and amino acid metabolism. Results of the metabolic evaluation in these rat brains were entirely consistent both with those illustrated in Figure 1, Figure 2, Figure 3, Figure 4 and Figure 5 and those described in a previous study [24], so that only modest effects on metabolism of 1 mg/kg b.w. ILB^®^ were observed, whilst a significant improvement of the same parameters was demonstrated with 5 mg/kg b.w. ILB^®^, compared to that found in untreated sTBI rats. Data from the neurocognitive function assessment clearly mirrored those of the metabolic evaluation, with remarkable neurocognitive deficits in untreated sTBI rats, no improvement induced by treatment with 1 mg/kg b.w. ILB^®^, and significant amelioration of all the parameters measured when 5 mg/kg b.w. ILB^®^ were administered 30 min post-sTBI.

In spite of the large number of papers published on TBI, few focused on the simultaneous measurement of metabolic and neurofunctional parameters, showing correlations between TBI-induced neurocognitive deficits and imbalance of brain metabolism [47,48,49,50,51,52]. In this model of sTBI, our results clearly evidenced that the profound metabolic imbalance of brain metabolism is accompanied by significant neurocognitive deficits and that, most importantly, the improvement of biochemical pathways and cycles (connected to mitochondrial functions, energy metabolism, antioxidant defenses and amino acid metabolism) induced by the administration of ILB^®^ provoked a concomitant amelioration of neurocognitive functions in sTBI-injured rats.

Although the exact mechanism of action of ILB^®^ has not yet been fully identified, there is evidence of the powerful antioxidant activity of different sulphated polysaccharides (SP), under various experimental conditions [53,54,55,56]. Further studies with ILB^®^ performed in human cells in vitro, in the rodent brain after sTBI, and in healthy humans and people with amyotrophic lateral sclerosis describe the remarkable anti-inflammatory activity of the drug, exploited through the biomodulation of different cytokines (e.g., TNF-α, IL-6 and TGF-β family) and growth factors (e.g., HGF, BDNF, VEGF) [57]. The results presented here and previously [24] clearly show that ILB^®^ treatment increases water-soluble antioxidants (ascorbate and GSH) and decreases oxidative/nitrosative stress. Since the involvement of both in sTBI-induced oxidative/nitrosative stress and neuroinflammation has been well established [20,21,24,58], it is conceivable to hypothesize that the benefits of ILB^®^ treatment on brain metabolism and neurocognitive functions following sTBI are mediated, at least in part, by the aforementioned antioxidant and anti-inflammatory activities of this drug. It is also worth mentioning that, in different models of cellular and organ toxicity, SP positively influences mitochondrial metabolism, with inhibition of apoptosis and ROS formation [59,60], and that, in a model of controlled cortical impact TBI, it decreases brain oxidative stress and mitochondrial dysfunction [61]. Altogether, this information supports the hypothesis that the effects of ILB^®^ administration, evidenced in the current and in previous studies [24,34,57], may be mediated by its anti-inflammatory and antioxidant activities that significantly attenuate mitochondrial dysfunction, recover the energy penalty, decrease Glu excitotoxicity, inhibit NO overproduction, restore the proper biochemical conditions for cerebral methylation reactions and normalize brain NAA homeostasis. These general metabolic ameliorations, primed by ILB^®^ administration, consequently caused evident improvement of neurocognitive functions of the sTBI-injured brain.

The results of the current and previous studies on sTBI [24] represent solid evidence for the utility of ILB^®^ to treat a range of acute brain sufferance with similar biochemical perturbations and, in particular, support planned future clinical studies aimed at showing the benefits of ILB^®^ treatment on the outcome of TBI patients.

## 4. Materials and Methods

### 4.1. Animals, TBI Induction and Drug Administration Protocol

The experimental study was approved by the Ethical Committee of the Catholic University of the Sacred Heart of Rome, Italy (approval 1F295.52, released on 20 October 2017), and by the Ethical Committee of the Italian Ministry of Health (approval No. 78/2018-PR, released on 5 February 2018). Male Wistar rats (*n* = 176) of 300–350 g body weight (b.w.) were maintained under controlled conditions and fed with a standard laboratory diet and water ad libitum. On the day of impact, animals were anesthetized with an intramuscular injection of a solution containing 35 mg/kg b.w. ketamine and 0.25 mg/kg b.w. midazolam. The “weight drop” impact acceleration model of TBI, set up by Marmarou A. et al. [62], was used to induce a diffused sTBI by dropping a 450 g weight from 2 m height onto the helmet protected rat head, in order to prevent skull fracture and distribute the impact force uniformly to the brain. Animals were not included in the study if manifesting skull fracture, seizures or nasal bleeding (instant and overall mortality rates of 7.7% and 29.5%, respectively, were recorded). Animals were sacrificed at 2 or 7 days post sTBI, immediately after a new anesthesia administration.

Nine groups of 12 rats each were obtained, as follows: controls = sham-operated rats, receiving neither sTBI nor drug treatment, sacrificed 48 h after anesthesia and all surgical procedures; 1 = sTBI only, sacrificed at 2 days post-injury; 2 = sTBI + 1 mg/kg b.w. ILB^®^, sacrificed at 2 days post-injury; 3 = sTBI + 5 mg/kg b.w. ILB^®^, sacrificed at 2 days post-injury; 4 = sTBI + 15 mg/kg b.w. ILB^®^, sacrificed at 2 days post-injury; 5 = sTBI only, sacrificed at 7 days post-injury; 6 = sTBI + 1 mg/kg b.w. ILB^®^, sacrificed at 7 days post-injury; 7 = sTBI + 5 mg/kg b.w. ILB^®^, sacrificed at 7 days post-injury; 8 = sTBI + 15 mg/kg b.w. ILB^®^, sacrificed at 7 days post-injury.

A stock solution of 20 mg/mL ILB^®^ (Tikomed AB, Viken, Sweden; batch 3045586) was diluted appropriately with sterile NaCl (9 mg/mL) and injected as a single subcutaneous administration (500 μL). The final amounts of drug, injected 30 min after sTBI, were 1, 5, and 15 mg/kg b.w. of ILB^®^.

In a different set of experiments, dedicated to evaluating the effects of ILB^®^ administration on neurocognitive functions of post-injured animals, rats were randomly assigned to four treatment groups, as follows: controls = sham-operated rats, receiving neither sTBI nor drug treatment, sacrificed 48 h after anesthesia and all surgical procedures; 1 = sTBI only, sacrificed at 7 days post-injury; 2 = sTBI + 1 mg/kg b.w. ILB^®^, sacrificed at 7 days post-injury; and 3 = sTBI + 5 mg/kg b.w. ILB^®^, sacrificed at 7 days post-injury. Injured animals underwent the same anesthetic, surgical, sTBI and drug administration protocols described above. All animals were studied over 7 days to assess their neurocognitive functions, as described in detail below. The rationale for eliminating from the neurocognitive study the highest ILB^®^ dose (15 mg/kg b.w.), used in the FAA study, derives from the recent clinical trials in which ILB^®^, administered at the concentration of 1 mg/kg b.w., ameliorated both serum biochemical markers and clinical symptoms of patients suffering from amyotrophic lateral sclerosis (ALS) [34,57]. Therefore, also in light of ethical considerations to keep to a minimum the number of animals used in this neurocognitive study, we selected the same ILB^®^ dose of these clinical trials (1 mg/kg b.w.) and a dose five times higher (5 mg/kg b.w.), for better adherence to the clinical settings. At the end of the final neurocognitive assessment session, animals were sacrificed, and the brains were removed and processed for the analysis of selected metabolites (ATP, GTP, NAD^+^, ascorbate, GSH, NAA, Glu, GABR, Met) according to the procedures described below.

### 4.2. Tissue Preparation and HPLC Analysis of Free Amino Acids, Energy Metabolites and Antioxidants

On the day of sacrifice (2 or 7 days post impact or, in the case of controls, 2 days after surgical procedures), animals were anesthetized and submitted to an in vivo craniotomy, as previously described in detail [16,17,18,19,20,21,22,23,24]. One hemisphere was freeze-clamped by aluminum tongue pre-cooled in liquid nitrogen and then immersed in liquid nitrogen, in order to minimize metabolite loss. After the wet weight (w.w.) determination of the tissue, organic solvent deproteinization was performed by adding appropriate amounts of ice-cold precipitating solution, composed of CH_3_CN (acetonitrile) + 10 mM KH_2_PO_4_, pH 7.40 (3:1; v:v), to the frozen tissue to obtain a 1:10 (w/v) homogenate [16,17,18,19,20,21,22,23,24]. An Ultra-Turrax homogenizer set at 24,000 rpm/min (Janke & Kunkel, Staufen, Germany) allowed the preparation of a fine homogenate in about 90 s. Tissue homogenates were then centrifuged at 20,690× *g* for 10 min at 4 °C, the clear supernatants were saved, pellets were supplemented with an aliquot of 10 mM KH_2_PO_4_, homogenized again as described above, and saved overnight at −20 °C in order to maximize extraction and recovery of water-soluble compounds from the brain tissue. These pellets were again centrifuged (20,690× *g* for 10 min at 4 °C), and the supernatants combined with those previously obtained. Removal of acetonitrile from the protein-free brain extracts was achieved by adding to clear supernatants a double volume of HPLC-grade CHCl_3_. After vortexing for 90 s, samples were centrifuged in a top-bench centrifuge at the maximal speed, the upper aqueous phase was removed, supplemented with the same chloroform volumes, vortexed and centrifuged again as above. Water-soluble low-molecular weight compounds, including FAA, were contained in the upper aqueous phases that were saved at −80 °C until the HPLC analysis of FAA.

A Surveyor HPLC system (Thermo Fisher Italia, Rodano, Milan, Italy) equipped with a highly sensitive photodiode array detector provided with a 5 cm light path flow cell, and set up between 200 and 400 nm wavelength, was used for the sample analysis. Data acquisition and analysis were performed using the ChromQuest^®^ software package, version 5.0 provided by the HPLC manufacturer (Thermo Fisher Scientific, Waltham, MA, USA).

FAA, including aspartate (Asp), glutamate (Glu), asparagine (Asn), serine (Ser), glutamine (Gln), histidine (His), glycine (Gly), threonine (Thr), citrulline (Cit), arginine (Arg), alanine (Ala), taurine (Tau), γ-aminobutyric acid (GABA), tyrosine (Tyr), S-adenosylhomocysteine (SAH), L-cystathionine (L-Cystat), valine (Val), methionine (Met), tryptophan (Trp), phenylalanine (Phe), isoleucine (Ile), leucine (Leu), ornithine (Orn) and lysine (Lys), plus the internal standard norvaline (Norval), were automatically derivatized before injection with a mixture of 25 mmol/L orthophthalaldehyde (OPA), 1% 3-methylpropionic acid (MPA) and 237.5 mmol/L sodium borate, pH 9.8, as described in detail elsewhere [63]. The automated precolumn derivatization of the samples with OPA-MPA was carried out at 25 °C, and 25 μL of the derivatized mixture were loaded onto a Hypersil C-18, 250 × 4.6 mm, 5 µm particle size HPLC column, thermostated at 21 °C during chromatographic runs. The flow rate was 1.2 mL/min, and a step gradient with two mobile phases (A = 24 mmol/L CH_3_COONa + 24 mmol/L Na_2_HPO_4_ + 1% tetrahydrofuran + 0.1% trifluoroacetic acid, pH 6.5; B = 40% CH_3_OH + 30% CH_3_CN + 30% H_2_O) allowed the separation of FAA of interest [23,63]. Assignment and calculation of derivatized FAA in whole brain extract runs were carried out at 338 nm wavelengths by comparing retention times and areas of peaks with those of chromatographic runs of freshly prepared ultra-pure standard mixtures containing known FAA concentrations.

Energy metabolites and water-soluble antioxidants were separated and quantified in the brain extracts of the neurocognitive study animal groups, as described in detail elsewhere [16,17,18,19,20,21,22,24].

### 4.3. Assessment of Neurocognitive Functions

All tests were performed during the first week after sTBI induction. On day 1 after injury, animals underwent the Open Field Test (OFT) for locomotor and anxiety assessment. This stage was also used as the habituation phase for both Novel Object Recognition (NOR) tests. On day 2, rats performed the training phase for the NOR-Long-Term Memory (NOR-LTM) test, followed on day 3 by the true test phase. On day 4 after injury, animals underwent both training and test phases for the NOR–Short-Term Memory (NOR-STM) test. All tests were performed by an experimenter blind to treatments.

For the OFT, animals were allowed to explore a 90 cm × 90 cm square arena for 10 min. The arena was divided into two zones: (1) a central zone, constituted by the inner square of (45 cm × 45 cm); (2) a border zone, constituted by the external squares. Time spent in both zones and distance traveled were recorded for the analysis. A higher portion of time spent in the central zone correlates with reduced anxiety, while higher time in the border zone correlates with higher anxiety. The arena was cleaned with 70% ethanol and water between tests of each other animal. Tracking analysis was performed using the latest software ANY-maze™ version (Stoelthing Co., Wood Dale, IL, USA).

The NOR-LTM test is divided into three phases: a habituation phase, a training phase and a test phase. The previous OFT was used as the habituation phase, in order to allow the animals to familiarize with the arena. Twenty-four hours after the habituation phase, two identical objects were placed symmetrically with respect to the center of the arena, and animals were allowed to explore the arena and the objects for 10 min (training phase). Time spent exploring these two objects was recorded. Twenty-four hours after the training phase, animals underwent the test phase: one of the two objects was removed and substituted with a different object. Animals were allowed to explore the arena and the objects for 5 min, and time spent exploring both objects was scored. Object exploration is defined as the animal’s snout pointing toward the object within a 3 cm radius from the object. The ratio between time spent exploring the novel object and total object exploration time (new object + old object) multiplied by 100 constitutes the preference index (PI). The same amount of time spent exploring both objects will result in a 50% PI, suggesting no discrimination and poor object memory, whilst a PI around 60% or higher suggests preference for the novel object and, therefore, intact object memory. Between each animal test, the arena and the object were cleaned with 70% ethanol to eliminate odors and avoid any potential confounding factors. Scoring of object exploration was performed using the latest software ANY-maze™ version (Stoelthing Co., Wood Dale, IL, USA).

The NOR-STM test was performed in the identical way to the NOR-LTM, except for the time intervals between the training and test phases and the duration of both phases. In the NOR-STM paradigm, the interval between training and test was 5 min, and the duration of both training and test was 3 min. A different pair of objects from those used in the NOR-LTM was chosen for this test to avoid potentially confounding factors. The NOR-STM was performed 24 h after the test phase of the NOR-LTM. Moreover, in this case, the scoring of object exploration was performed using the latest software ANY-maze™ version (Stoelthing Co., Wood Dale, IL, USA)

### 4.4. Statistics

Statistical analysis was performed using the GraphPad Prism program, 8.01 version (GraphPad Software, San Diego, CA 92108, USA). Continuous variables were expressed as mean ± SD. Since not all data displayed normal distributions, tested according to the Kolmogorov–Smirnov test, differences among groups were determined by the Kruskal–Wallis non-parametric tests for multiple comparisons, corrected by controlling the false discovery rate (FDR) using the two-stage linear step-up procedure of Benjamini, Krieger and Yekutieli. Differences were considered significant when *q* < 0.05.

## Figures and Tables

**Figure 1 ijms-23-08460-f001:**
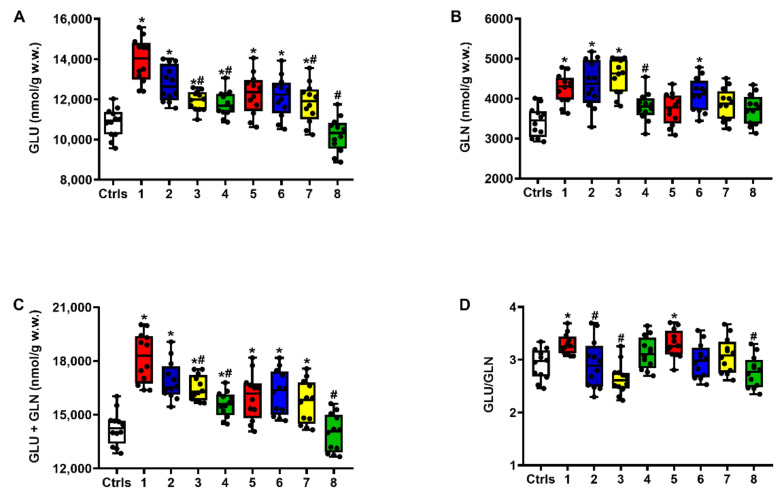
The box plots reporting all data points (•), minimum, maximum, median, 25 and 75 percentiles of Glu (**A**), Gln (**B**) and Glu + Gln (**C**), as well as the values of Glu/Gln ratio (**D**), in brain extracts of sTBI rats without and with the s.c. administration of increasing ILB^®^ doses (1, 5 and 15 mg/kg b.w.), measured at 2 and 7 days post-impact. Ctrls = control sham-operated rats; 1 = sTBI only, sacrificed at 2 days post-injury; 2 = sTBI + 1 mg/kg b.w. ILB^®^, sacrificed at 2 days post-injury; 3 = sTBI + 5 mg/kg b.w. ILB^®^, sacrificed at 2 days post-injury; 4 = sTBI + 15 mg/kg b.w. ILB^®^, sacrificed at 2 days post-injury; 5 = sTBI only, sacrificed at 7 days post-injury; 6 = sTBI + 1 mg/kg b.w. ILB^®^, sacrificed at 7 days post-injury; 7 = sTBI + 5 mg/kg b.w. ILB^®^, sacrificed at 7 days post-injury; 8 = sTBI + 15 mg/kg b.w. ILB^®^, sacrificed at 7 days post-injury. * Significantly different from controls, *q* < 0.05. # Significantly different from corresponding time of untreated sTBI rats, *q* < 0.05.

**Figure 2 ijms-23-08460-f002:**
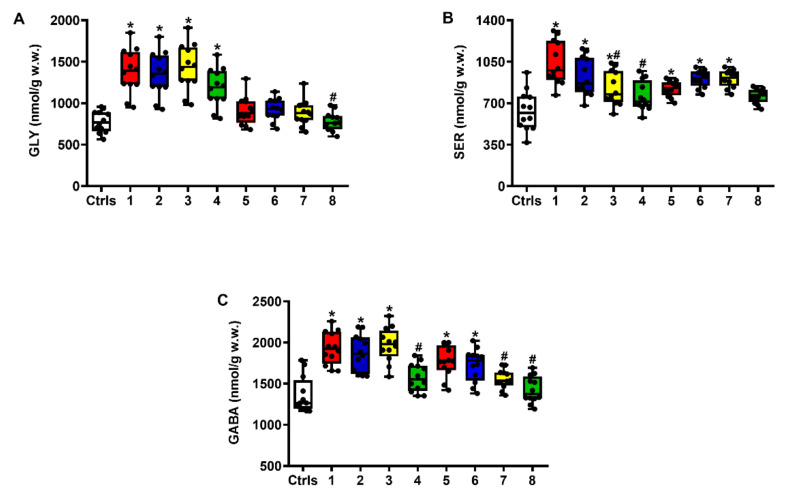
The box plots reporting all data points (•), minimum, maximum, median, 25 and 75 percentiles of Gly (**A**) and Ser (**B**) and GABA (**C**) in rats without and with the s.c. administration of increasing ILB^®^ doses (1, 5 and 15 mg/kg b.w.), measured at 2 and 7 days post-impact. Ctrls = control sham-operated rats; 1 = sTBI only, sacrificed at 2 days post-injury; 2 = sTBI + 1 mg/kg b.w. ILB^®^, sacrificed at 2 days post-injury; 3 = sTBI + 5 mg/kg b.w. ILB^®^, sacrificed at 2 days post-injury; 4 = sTBI + 15 mg/kg b.w. ILB^®^, sacrificed at 2 days post-injury; 5 = sTBI only, sacrificed at 7 days post-injury; 6 = sTBI + 1 mg/kg b.w. ILB^®^, sacrificed at 7 days post-injury; 7 = sTBI + 5 mg/kg b.w. ILB^®^, sacrificed at 7 days post-injury; 8 = sTBI + 15 mg/kg b.w. ILB^®^, sacrificed at 7 days post-injury. * Significantly different from controls, *q* < 0.05. # Significantly different from corresponding time of untreated sTBI rats, *q* < 0.05.

**Figure 3 ijms-23-08460-f003:**
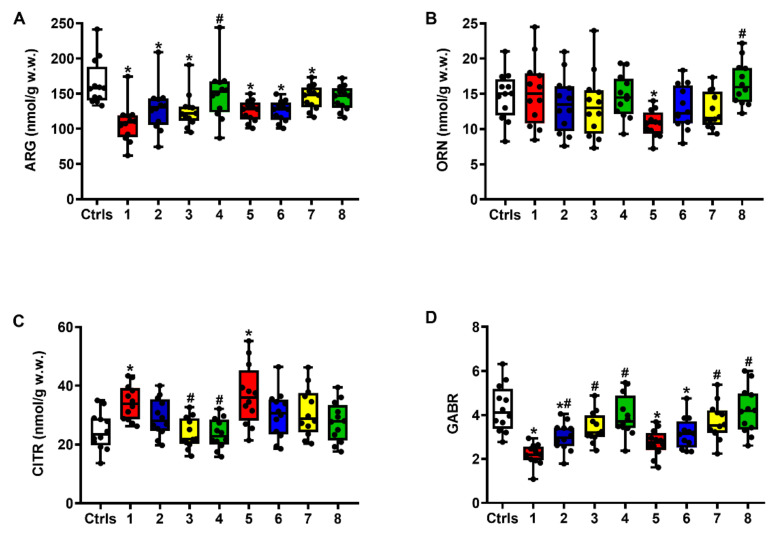
The box plots reporting all data points (•), minimum, maximum, median, 25 and 75 percentiles of Arg (**A**) and its metabolites Orn (**B**) and Citr (**C**), together with the global arginine bioavailability ratio (GABR = Arg/Orn + Citr) (**D**), in rats without and with the s.c. administration of increasing ILB^®^ doses (1, 5 and 15 mg/kg b.w.). Ctrls = control sham-operated rats; 1 = sTBI only, sacrificed at 2 days post-injury; 2 = sTBI + 1 mg/kg b.w. ILB^®^, sacrificed at 2 days post-injury; 3 = sTBI + 5 mg/kg b.w. ILB^®^, sacrificed at 2 days post-injury; 4 = sTBI + 15 mg/kg b.w. ILB^®^, sacrificed at 2 days post-injury; 5 = sTBI only, sacrificed at 7 days post-injury; 6 = sTBI + 1 mg/kg b.w. ILB^®^, sacrificed at 7 days post-injury; 7 = sTBI + 5 mg/kg b.w. ILB^®^, sacrificed at 7 days post-injury; 8 = sTBI + 15 mg/kg b.w. ILB^®^, sacrificed at 7 days post-injury. * Significantly different from controls, *q* < 0.05. # Significantly different from corresponding time of untreated sTBI rats, *q* < 0.05.

**Figure 4 ijms-23-08460-f004:**
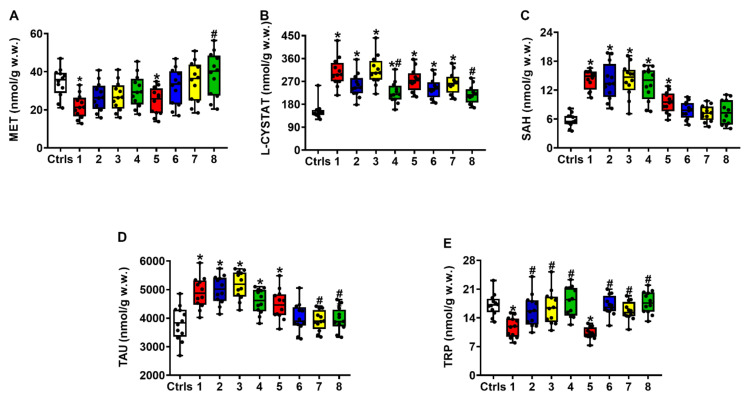
The box plots reporting all data points (•), minimum, maximum, median, 25 and 75 percentiles of Met (**A**), L-Cystat (**B**), SAH (**C**), Tau (**D**) and Trp (**E**), in rats without and with the s.c. administration of increasing ILB^®^ doses (1, 5 and 15 mg/kg b.w.). Ctrls = control sham-operated rats; 1 = sTBI only, sacrificed at 2 days post-injury; 2 = sTBI + 1 mg/kg b.w. ILB^®^, sacrificed at 2 days post-injury; 3 = sTBI + 5 mg/kg b.w. ILB^®^, sacrificed at 2 days post-injury; 4 = sTBI + 15 mg/kg b.w. ILB^®^, sacrificed at 2 days post-injury; 5 = sTBI only, sacrificed at 7 days post-injury; 6 = sTBI + 1 mg/kg b.w. ILB^®^, sacrificed at 7 days post-injury; 7 = sTBI + 5 mg/kg b.w. ILB^®^, sacrificed at 7 days post-injury; 8 = sTBI + 15 mg/kg b.w. ILB^®^, sacrificed at 7 days post-injury. * Significantly different from controls, *q* < 0.05. # Significantly different from corresponding time of untreated sTBI rats, *q* < 0.05.

**Figure 5 ijms-23-08460-f005:**
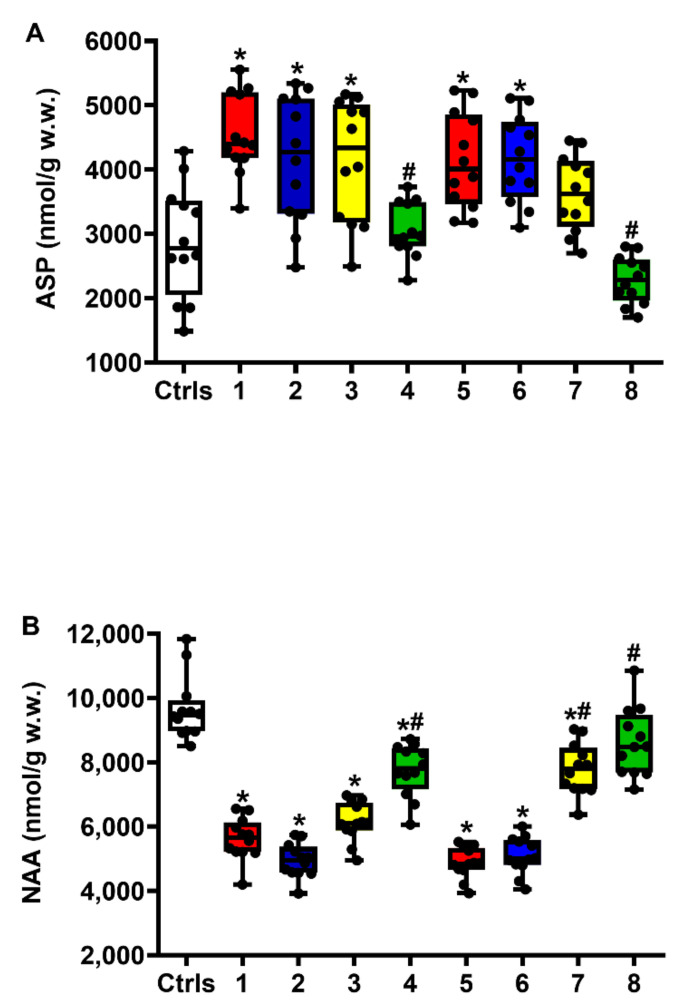
The box plots reporting all data points (•), minimum, maximum, median, 25 and 75 percentiles of Asp (**A**) and NAA (**B**) in rats without and with the s.c. administration of increasing ILB^®^ doses (1, 5 and 15 mg/kg b.w.). Ctrls = control sham-operated rats; 1 = sTBI only, sacrificed at 2 days post-injury; 2 = sTBI + 1 mg/kg b.w. ILB^®^, sacrificed at 2 days post-injury; 3 = sTBI + 5 mg/kg b.w. ILB^®^, sacrificed at 2 days post-injury; 4 = sTBI + 15 mg/kg b.w. ILB^®^, sacrificed at 2 days post-injury; 5 = sTBI only, sacrificed at 7 days post-injury; 6 = sTBI + 1 mg/kg b.w. ILB^®^, sacrificed at 7 days post-injury; 7 = sTBI + 5 mg/kg b.w. ILB^®^, sacrificed at 7 days post-injury; 8 = sTBI + 15 mg/kg b.w. ILB^®^, sacrificed at 7 days post-injury. * Significantly different from controls, *q* < 0.05. # Significantly different from corresponding time of untreated sTBI rats, *q* < 0.05.

**Figure 6 ijms-23-08460-f006:**
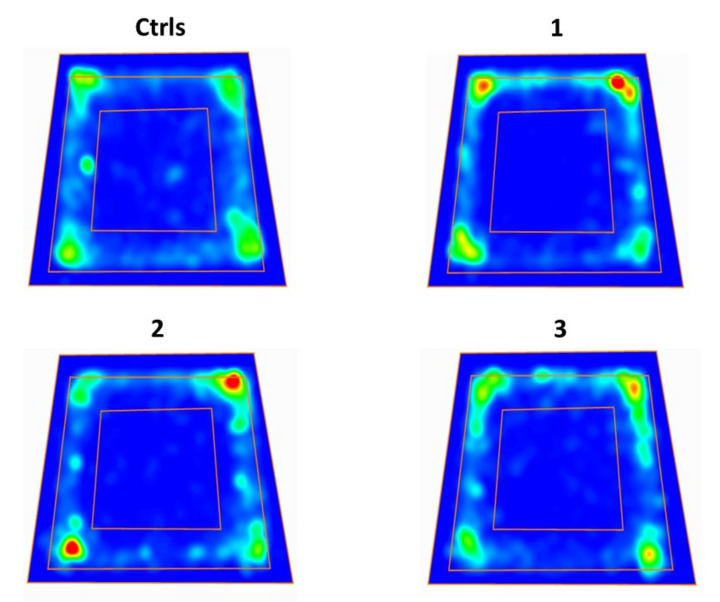
Heat maps showing the mean time spent by each treatment group in the central and peripheral zones of the test arena, where the red color indicates the maximum time of occupancy in either zone (=30 s). Ctrls = control sham-operated rats; 1 = sTBI only, sacrificed at 7 days post-injury; 2 = sTBI + 1 mg/kg b.w. ILB^®^, sacrificed at 7 days post-injury; 3 = sTBI + 5 mg/kg b.w. ILB^®^, sacrificed at 7 days post-injury.

**Figure 7 ijms-23-08460-f007:**
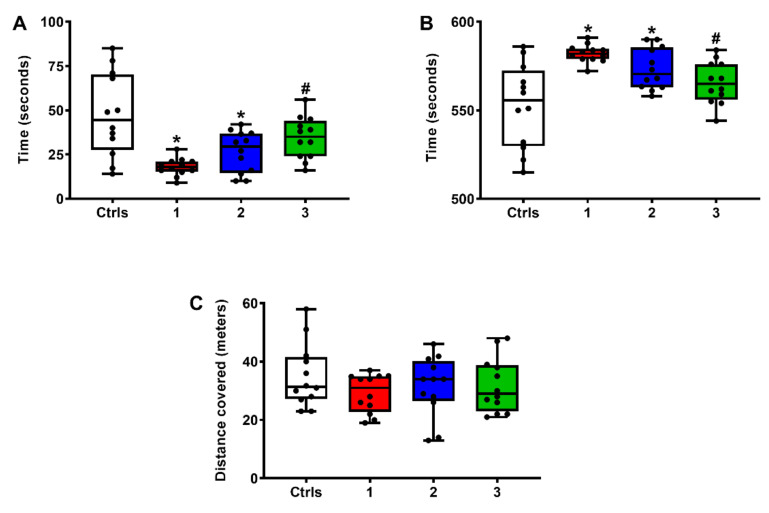
The box plots reporting all data points (•), minimum, maximum, median, 25 and 75 percentiles of the time (in seconds) spent in the center (**A**) or in peripheral (**B**) zones and the distance traveled (**C**) in the test arena (open field test, OFT) at 24 h post-injury by rats without and with the s.c. administration of two ILB^®^ doses (1 and 5 mg/kg b.w.). Ctrls = control sham-operated rats; 1 = sTBI only, sacrificed at 7 days post-injury; 2 = sTBI + 1 mg/kg b.w. ILB^®^, sacrificed at 7 days post-injury; 3 = sTBI + 5 mg/kg b.w. ILB^®^, sacrificed at 7 days post-injury. * Significantly different from controls, *q* < 0.05. # Significantly different from untreated sTBI rats, *q* < 0.05.

**Figure 8 ijms-23-08460-f008:**
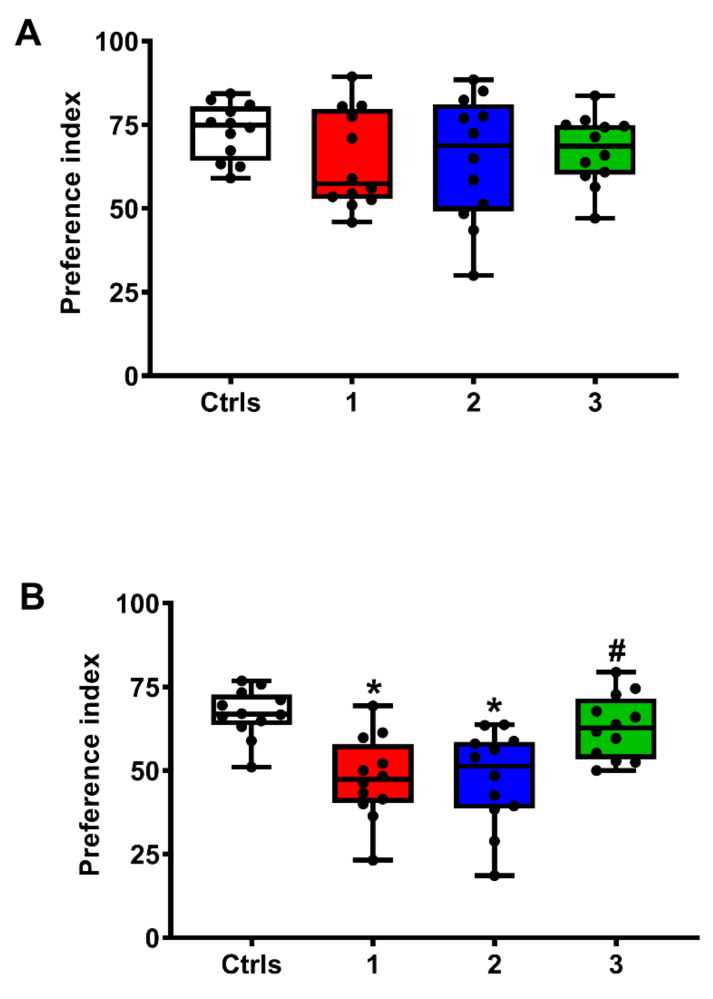
The box plots reporting all data points (•), minimum, maximum, median, 25 and 75 percentiles of the Preference Index (PI), as a measure of NOR-STM (**A**) and NOR-LTM (**B**), at 7 days post-injury of rats without and with the s.c. administration of two ILB^®^ doses (1 and 5 mg/kg b.w.). Ctrls = control sham-operated rats; 1 = sTBI only, sacrificed at 7 days post-injury; 2 = sTBI + 1 mg/kg b.w. ILB^®^, sacrificed at 7 days post-injury; 3 = sTBI + 5 mg/kg b.w. ILB^®^, sacrificed at 7 days post-injury. * Significantly different from controls, *q* < 0.05. # Significantly different from untreated sTBI rats, *q* < 0.05.

**Table 1 ijms-23-08460-t001:** Concentrations of selected cerebral metabolites determined in deproteinized brain extracts of rats undergoing the neurocognitive study. Control rats received surgical procedures and not sTBI, whilst the other three groups experienced sTBI. Those treated with ILB^®^ received a subcutaneous injection of the drug (at the two dose levels indicated in the table), 30 min after sTBI induction. Animals were sacrificed 7 days post-sTBI or surgical procedure (in the case of sham operated controls).

nmol/g w.w.	Controls	sTBI	sTBI + ILB^®^ 1 mg/kg b.w.	sTBI + ILB^®^ 5 mg/kg b.w.
ATP	2278 ± 356	1472 ± 226 ^a^	1689 ± 193 ^a^	1805 ± 237 ^a,b^
GTP	621 ± 76	401 ± 59 ^a^	452 ± 63 ^a^	511 ± 70 ^a,b^
NAD^+^	542 ± 39	304 ± 44 ^a^	296 ± 38 ^a^	395 ± 48 ^a,b^
Ascorbate	3005 ± 412	2126 ± 378 ^a^	2314 ± 404 ^a^	2456 ± 512 ^a^
GSH	3656 ± 592	2014 ± 194 ^a^	2166 ± 388 ^a^	2704 ± 259 ^a,b^
NAA	9756 ± 753	5031 ± 642 ^a^	5389 ± 703 ^a^	6976 ± 775 ^a,b^
Glu	10358 ± 1202	12548 ± 1311 ^a^	12001 ± 1154 ^a^	11782 ± 1096 ^a,b^
GABR	3.94 ± 0.91	2.55 ± 0.55 ^a^	2.89 ± 0.78 ^a^	3.21 ± 0.85 ^b^
Met	42.50 ± 7.33	28.72 ± 6.02 ^a^	33.14 ± 5.70	36.41 ± 6.95

Values are the mean ± S.D. of 12 different animals. GABR = global arginine bioavailability ratio = Arg/Orn + Citr. ^a^ Significantly different from sham operated controls, *q* < 0.05. ^b^ Significantly different from untreated sTBI rats, *q* < 0.05.

## Data Availability

The data presented in this study are available from the Figure 1, Figure 2, Figure 3, Figure 4, Figure 5, Figure 6, Figure 7 and Figure 8, where all data points are clearly indicated. Raw data can be obtained on request to corresponding Authors (S.S. and B.T.).

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
