# Peer review of "ILB®, a Low Molecular Weight Dextran Sulphate, Restores Glutamate Homeostasis, Amino Acid Metabolism and Neurocognitive Functions in a Rat Model of Severe Traumatic Brain Injury"

_ijms, 2022, doi:10.3390/ijms23158460_

Round 1

Reviewer 1 Report

In the previous study, the authors performed an experimental study observing the beneficial effect of ILB in a rat model of severe traumatic brain injury. It was shown that ILB was able to improve brain energy metabolisms (ATP/ADP) and mitochondrial capacity with decrease of oxidative stress. In the present study, they analyzed further details concerning Glu/Gyl excitoxicity, metabolites and neurocognitive functions. All in all, ILB decreased whole brain glutamate, glycine, serine and 51-aminobutyric acid when ILB was given at least 5 or 15mg/kg s.c.  The beneficial effect could be observed in the neurocognitive function analysis as well. To sum up, the authors conclude ILB medication as a potential drug for the treatment of severe TBI. 

In my opinion, the current study is plausible and well done. In the future study, animal MRIs with illustration of diffuse axonal injury might give the study more impact and clinical relevance. 

I have only minor issues to comment:

Introduction:

-The introduction is too long. In the previous study, the authors reported the background in detail. In the current study, I think that the introduction could be more focused and shortened. 

Methods:

-The authors analyzed 5 groups a 12 rats. A previously published trauma model by Marmarou et al. was applied for this study.  He reported a mortality of 44% whereas the authors had only a low overall mortality of 7.7%. Furthermore, animals with skull fracture or seizure were not included. How was the diagnosis of skull fracture made? If seizure occurred during the 7 days period, were the animals excluded? Why were the rats excluded if they had a seizure (With the background that the risk of seizure is high in severe TBI)? The animals with additional seizure might have been interesting to analyze separately. How many animals were excluded in total?

-sham-operated rats were sacrificed 7days after all surgical procedures. However in the section 4.2, the author described that the case of controls were sacrificed 2days after surgical procedures. 

-Minor grammatical errors to address:

“at2 or 7 days postsTBI“ (4.1), “ofCH3CN” (4.2), 

Results:

-Figure 6: TBI +15mg/kg ILB?  - I thought the dosage was 5mg/kg ILB. Please clarify

-Figure 6: I think that all the rats spent higher time in the border zone. The difference what I noticed is that rats with sTBI and sTBI+1mg/kg spent most of the time in one spot whereas the rat with sTBI+5mg/kg spent in several spots in the border zone. Please explain.

-Figure7: There was no significant difference concerning covered distance between those groups. The heatmap from Figure 6 is telling us different result?

Discussions:

-“The recent positive results of ILB® treatment in two clinical trials in ALS patients [33, 492 40, 63] and the results of the current and previous studies on sTBI [30], represent solid evidence for the utility of ILB® to treat a range of acute and chronic neurological conditions with similar biochemical perturbations and, in particular, support planned future clinical studies aimed to show the benefits of ILB® treatment on the outcome of TBI patients”

-- It is interesting to know that ILB treatment shows positive results in ALS patients, however, this information is not helping us to understand the effect of ILB in patients with sTBI. The author conclude that ILB could be treated in a range of acute and chronic neurological conditions. That statement can only be made if the author go into detail concerning the effect of ILB in patients with ALS. Otherwise, I would exclude the statement and only focus on acute neurological conditions.

Author Response

File uploaded with response to Reviewers.

Reviewer 2 Report

Review of manuscript entilted: “ILB®, a low molecular weight dextran sulphate,restores glutamate homeostasis, amino acid metabolism and neurocognitive functions in a rat model of severe traumatic brain injury” authored by Giacomo Lazzarino, Valentina Di Pietro, Marco Rinaudo, Zsuzsanna Nagy, Nicholas M. Barnes, Lars Bruce, Stefano Signoretti, Renata Mangione, Miriam Wissam Saab, Barbara Tavazzi, Antonio Belli, Giuseppe Lazzarino, Angela Maria Amorini and Ann Logan 

In the presented manuscript authors investigated effects of new compound, which is supposed to attenuate severe traumatic brain injury (sTBI) negative outcomes. Introduction provides valuable information and justifies undertaken problem. Methods are described in detailed manner. Results have to be presented more clearly, since graphs are not reader-friendly. Discussion is written logically and based on obtained results but I have one remark, which may be worth discussing.

Overall manuscript is well-written and touches very important problem. Just because of my curiosity and scientific field, I would like to ask you about potential influence of ILB® on the blood-brain barrier functional state (since you mentioned anti-inflammatory properties of your compound). Moreover, it is proven that TBI leads to blood-brain barrier breakdown, which furtherly may cause compensatory overproduction of particular neurotransmitters due to their escape from central nervous system. This topic could be added to discussion of your work and may be promising target for further research.

Major concerns:

  • Graphs – x-axis, please try to make it more clear, since in the present version it is hard to read all the headers.

Minor concerns:

  • Figure 6. – there is an 15mg/kg dose and the TBI (not sTBI) is it an additional group which is not mentioned in the text?

Author Response

(The authors gave the same response as above.)
